# Preceding Phenological Events Rather than Climate Drive the Variations in Fruiting Phenology in the Desert Shrub *Nitraria tangutorum*

**DOI:** 10.3390/plants11121578

**Published:** 2022-06-15

**Authors:** Fang Bao, Zhiming Xin, Minghu Liu, Jiazhu Li, Ying Gao, Qi Lu, Bo Wu

**Affiliations:** 1Institute of Desertification Studies, Chinese Academy of Forestry, Beijing 100091, China; leejzids@caf.ac.cn (J.L.); ying_gao16@163.com (Y.G.); luqi@caf.ac.cn (Q.L.); 2Institute of Ecological Conservation and Restoration, Chinese Academy of Forestry, Beijing 100091, China; 3Key Laboratory for Desert Ecosystem and Global Change, Chinese Academy of Forestry, Beijing 100091, China; 4Gansu Minqin Desert Ecosystem National Observation Research Station, Wuwei 733300, China; 5Experimental Center of Desert Forestry, Chinese Academy of Forestry, Bayannaoer 015200, China; xzmlkn@163.com (Z.X.); slzxlmh@sina.com (M.L.)

**Keywords:** desert species, fruit setting, fruit ripening, water addition, phenology

## Abstract

Fruit setting and ripening are crucial in the reproductive cycle of many desert plant species, but their response to precipitation changes is still unclear. To clarify the response patterns, a long-term in situ water addition experiment with five treatments, namely natural precipitation (control) plus an extra 25%, 50%, 75%, and 100% of the local mean annual precipitation (145 mm), was conducted in a temperate desert in northwestern China. A whole series of fruiting events including the onset, peak, and end of fruit setting and the onset, peak, and end of fruit ripening of a locally dominant shrub, *Nitraria tangutorum*, were observed from 2012 to 2018. The results show that (1) water addition treatments had no significant effects on all six fruiting events in almost all years, and the occurrence time of almost all fruiting events remained relatively stable compared with leaf phenology and flowering phenology after the water addition treatments; (2) the occurrence times of all fruiting events were not correlated to the amounts of water added in the treatments; (3) there are significant inter-annual variations in each fruiting event. However, neither temperature nor precipitation play key roles, but the preceding flowering events drive their inter-annual variation.

## 1. Introduction

Fruiting phenology includes the initiation, growth, and ripening of fruit [1]. It influences many aspects of ecosystem functioning such as insects laying eggs in fruits [2], bird–fruit associations [3], frugivorous animals feeding on fleshy fruits [4,5], and forest regeneration and dynamics [6], etc. High interspecific variation in the mean date of fruit ripening has been reported in tropical forests [4,7,8,9,10,11], temperate forests [7,12,13], and grassland [14,15,16,17,18], and these results are based mainly on tree species with their fruits as food sources of frugivores [4,5,19]. Despite their importance, less attention has been paid to the patterns and drivers of fruiting phenology in temperate desert ecosystems worldwide due to the sparseness of phenology records.

*Nitraria tangutorum* is a pioneer species widely distributed in the northwestern regions of China. It has a high tolerance to drought, high temperature, cold, salinity, alkalinity, bareness, wind erosion, and sand burial and plays a pivotal role in the desert ecology by fixing sands as well as slowing down deserts’ expansion. The fruit of *N. tangutorum*, called “desert cherry” (Figure 1C), is a native folk medicine used to treat dyspepsia, neurasthenia, abnormal menstruation, and heart disease in western China [20,21,22]. Furthermore, the berries provide *Eremias multiocellata* and *Eremias przewalskii* with the water they need during the summer heat [23]. Moreover, another valuable medicine, *Cynomorium songaricum* Rupr., which is parasitic on the roots of *N. tangutorum*, has a relevant socioeconomic importance that determines the management of *N. tangutorum*. Precipitation at the present study area has shown an increasing trend over the last 40 years (Figure 2A), matching a regional trend in northwestern China identified in [24,25,26]. The precipitation increase is causing the advance of leaf unfolding, the postponement of leaf cessation, and the prolongation of the growing season in *N. tangutorum* [27,28]. However, we know little about whether the fruiting phenology is altered. Characterizing the impacts of precipitation increase on the fruiting phenology of *N. tangutorum* is crucial given its ecological and economical importance. Here, we conducted a seven-year in situ simulated water addition experiment to evaluate the response of fruiting phenology to precipitation increase. Five water addition treatments were designed to simulate precipitation increase: natural precipitation (Control) and natural precipitation plus an extra 25% (+25%), 50% (+50%), 75% (+75%), and 100% (+100%) of the local annual mean precipitation (145 mm, 1978–2008). The whole series of the phenological events of *N. tangutorum*, including leaf unfolding (onset and end), flowering (onset and end), fruit setting (onset, peak, and end), fruit ripening (onset, peak, and end), and leaf cessation (leaf coloring and leaf fall), was detected. The aims of this study were to address the following specific questions: (1) How will water increase affect the patterns of fruiting phenology? (2) Do fruiting events shift consistently with the other phenological events? (3) Does water drive the variation in fruiting time?

## 2. Results

### 2.1. Inter-Annual Dynamics of Meteorological Factors

The climate at the study site is markedly seasonal, with a wet/warm season from May to September and a dry/cold season occurring from October to April. The annual precipitation at the study site showed an increasing pattern in the past 40 years (Figure 1A). The annual mean temperature (AMT), winter mean temperature (T_Win_, December to February), spring mean temperature (T_Spr_, March to May), summer mean temperature (T_Sum_, June to August), and autumn mean temperature (T_Aut_, September to November) all showed no significant increasing or decreasing trends (Bao et al., 2020). The annual mean precipitation (AMP) in 2012, 2013, 2014, 2015, 2016, 2017, and 2018 was 213.3, 59.1, 95.2, 147, 189.2, 86.0, and 59.1 mm, respectively. Winter rainfall was scarce (P_Win_, December to February), occurring only in three of the seven years, and the amounts were very low, with less than 5 mm (Bao et al., 2020). Spring (P_Spr_, March to May), summer (P_Sum_, June to August), and autumn (P_Aut_, September to November) rainfall varied from 2012 to 2018 with no increasing or decreasing patterns (Bao et al., 2020). This study focused on the variations in the accumulated precipitation and temperature prior to and during the fruiting period (1 January–31 August). The accumulated precipitation in 2012, 2013, 2014, 2015, 2016, 2017, and 2018 was 192.3, 51.5, 51.4, 80.9, 160.5, 63.8, and 56.2 mm, respectively. More than 95% of the rain fell during the fruiting period (May to August). These amounts are very close to the annual precipitation. Based on the deviation from the mean value of the seven years, 2012 and 2016 were “above-average” (i.e., wet) years; 2013 was an “ultra below-average” (i.e., extremely dry) year; 2014, 2017, and 2018 were “below-average” (i.e., dry) years; and 2015 was an “average” year (Figure 2B).

### 2.2. Changes in Fruit Setting

The linear mixed model analysis showed that the water addition treatments had no significant effects on the onset (OFS), peak (PFS), and end time (EFS) of fruit setting (all *p* > 0.05, Table 1) over the seven years (2012–2018). The year affected all three events significantly (all *p* < 0.05, Table 1) over the seven years, and there was no interaction between water addition treatment and year (all *p* > 0.05, Table 1).

The shifting direction of fruit setting events was not consistent among water addition treatments and among years in the 2012–2018 period (Figure 3).

The occurrence time of OFS was advanced in three (2012, 2013, and 2015) and delayed in one (2014) of the seven years under all four water addition treatments (+25%, +50%, +75%, and +100%) (all *p* > 0.05 with the exception of the +50% water addition treatment in 2015, Figure 3A). The shifting directions were not consistent among water addition treatments in the other years (Figure 3A). On average, over the seven years (2012–2018), the occurrence of OFS in the +25%, +50%, +75%, and +100% water addition treatment plots was not significantly changed compared with the control (all *p* > 0.05, Figure 3A, last column).

The occurrence time of PFS was advanced in one (2012) and delayed in one (2014) of the seven years under all four water addition treatments over the seven years (2012–2018) (all *p* > 0.05, Figure 3B). The shifting directions were not consistent among water addition treatments in the other years (Figure 3B). On average, over the seven years (2012–2018), the occurrence of PFS in +25%, +50%, +75%, and +100% water addition treatment plots were not significantly changed compared with control (all *p* > 0.05, Figure 3B, last column).

The occurrence time of EFS was advanced in two (2012 and 2018) and delayed in one (2014) of the seven years under all four water addition treatments over the seven years (2012–2018) (all *p* > 0.05 with the exception of the +50% and +100% water addition treatments in 2012, Figure 3C). The shifting directions were not consistent among water addition treatments in the other years (Figure 3B). On average, over the seven years (2012–2018), the occurrence of EFS in the +25%, +50%, +75%, and +100% water addition treatment plots was not significantly changed compared with the control (all *p* > 0.05, Figure 3C, last column).

### 2.3. Changes in Fruit Ripening

The linear mixed model analysis showed that the water addition treatments had no significant effects on the onset (OFR), peak (PFR), and end time (EFR) of fruit ripening (all *p* > 0.05, Table 1) over the seven years (2012–2018). The year affected all three events significantly (all *p* < 0.05, Table 1) over the seven years, and there was no interaction between water addition treatment and year (all *p* > 0.05, Table 1).

The shifting directions of fruit ripening events were not consistent among water addition treatments and among years over 2012–2018 (Figure 4).

There were no significant effects of water addition treatments on the occurrence time of OFR in six of the seven years, except 2012 (all *p* > 0.05, Figure 4A). On average, in the seven years (2012–2018), the occurrence of OFR in the +25%, +50%, +75%, and +100% water addition treatment plots was not significantly changed compared with the control (all *p* > 0.05, Figure 4, last column).

The occurrence time of PFR was advanced in 2012 under all four water addition treatments (all *p* < 0.05, Figure 4B). However, it was delayed in 2016 under all four water addition treatments (all *p* < 0.05, Figure 4B). There were no significant effects of water addition treatments on the occurrence time of PFR in the other five years from 2012 to 2018 (all *p* > 0.05, Figure 4B). On average, over the seven years (2012–2018), the occurrence of OFR in the +25%, +50%, +75%, and +100% water addition treatment plots was not significantly changed compared with the control (all *p* > 0.05, Figure 4B, last column).

The occurrence time of EFR was delayed in six of the seven years under all four water addition treatments, with the exception in 2012 (all *p* > 0.05, Figure 4C). On average, over the seven years (2012–2018), the occurrence time of EFR in the +25%, +50%, +75%, and +100% water addition treatment plots was not significantly changed compared with the control (all *p* > 0.05, Figure 4, last column).

### 2.4. Changes in the Duration of Fruiting Period

The linear mixed model analysis showed that the water addition treatments had no significant effects on the duration of the fruiting period (DF) (*p* > 0.05, Table 1) over the seven years (2012–2018). The year had a significant effect on it (*p* < 0.05, Table 1). There was no interaction effect between water addition treatment and year (*p* > 0.05, Table 1).

There were no significant effects of water addition treatment on the DF in six of the seven years (all *p* > 0.05, Figure 5), with the exception in 2016 under all four water addition treatments. The DF was significantly prolonged by 7.50, 4.00, 10.00, and 7.00 days by the +25%, +50%, +75%, and +100% treatments, respectively, in 2016 (*p* < 0.05 with the exception of the +50% treatment, Figure 5). On average, over the seven years, the DF was significantly prolonged by 3.32, 3.29, 3.29, and 3.14 days in the +25%, +50%, +75%, and +100% water addition treatments, respectively (*p* < 0.05, Figure 5).

### 2.5. Comparison with Other Phenological Events

On average, over the seven years (2012–2018), the water addition treatments significantly advanced the leaf unfolding and flowering events and delayed the leaf cessation events. For example, the end of leaf unfolding time (ELF) in the +25, +50, +75, and +100% water addition treatments was advanced by 5.07, 9.29, 9.29, and 7.86 d, respectively. The end of leaf coloring time (ELC) in the +25, +50, +75, and +100% water addition treatments was delayed by 2.32, 6.00, 9.07, and 12.32 d, respectively. Compared with leaf phenology, the variation in flower phenology was lower. The OFL in the +25, +50, +75, and +100% water addition treatments was significantly advanced by 0.43, 1.29, 2.43, 1.14 d, respectively (all *p* < 0.05). The EFL in the +25%, +50%, +75%, and +100% water addition treatments was significantly advanced by 1.89, 1.46, 2.25, and 1.14 d, respectively (all *p* < 0.05). By contrast, the changes in fruiting events were even smaller compared with the flowering events. We found that the responses of all fruiting events to the water addition treatments were relatively stable compared to the leaf and flowering phenology (Figure 5). The effects of the water addition treatments on fruiting events, namely OFS, PFS, EFS, OFR, PFR, and EFR, were not significant (Figure 6, all *p* > 0.05).

### 2.6. The Correlations between Fruiting Events and Water Addition Amounts

A simple linear regression analysis showed that there were no significant correlations between most fruiting events and the water addition amounts in almost all years (all *p* > 0.05), with the exceptions of OFS in 2012, 2014, and 2015; PFR in 2012; and EFR in 2018 (Figure 7). For the individual fruiting event OFS, which was obviously affected by the water addition treatments in three of the seven years, its response patterns to the water addition treatments among the three years were not consistent. A positive quadratic relationship in 2014 and negative quadratic relationships in 2012 and 2015 were determined (Figure 7A).

### 2.7. The Correlations between Fruiting Events and Water Climatic Factors

The inter-annual changes in all six fruiting events (OFS, PFS, EFS, OFR, PFR, and EFR) were correlated to neither the precipitation factors (AP, P_Win_, P_Spr_, P_Sum_, and P_Aut_) nor the temperature factors (AMT, T_Win_, T_Spr_, T_Sum_, and T_Aut_) under all treatments (Ctrl, +25%, +50%, +75%, and +100%) over the seven-year period (2012–2018) (all *p* > 0.05, Table 2 and Table 3).

### 2.8. The Correlations between Fruiting Events and the Other Phenological Events

The inter-annual changes in all six fruiting events (OFS, PFS, EFS, OFR, PFR, and EFR) were not correlated to any of the early-season leaf phenological events, OLU or ELU, under all water addition treatments (Ctrl, +25%, +50%, +75%, and +100%) over the seven-year period (2012–2018) (all *p* > 0.05). By contrast, we found that their inter-annual changes were almost all significantly correlated to the preceding flowering events, both OLF and ELF, under all treatments (Ctrl, +25%, +50%, +75%, and +100%) over the seven-year period (2012–2018) (Figure 8).

In Table 2 and Table 3, AMP and AMT represent the annual mean precipitation and annual mean temperature, respectively; T_Win_, T_Spr_, T_Sum_, and T_Aut_ represent the mean temperature during winter (December to February), spring (March to May), summer (June to August), and autumn (September to November), respectively; P_Win_, P_Spr_, P_Sum_, and P_Aut_ represent the accumulated precipitation during winter (December to February), spring (March to May), summer (June to August), and autumn (September to November), respectively; and OFS, PFS, and EFS represent the onset, peak, and end of the fruit setting period, respectively.

## 3. Discussion

### 3.1. Effects of Water Addition Treatments on Fruiting Events

Plants that live in deserts have to contend with stressful conditions characterized by sandy soils with low moisture and low fertility. These species are aridity tolerators and are also water-sensitive. The leaf unfolding and leaf coloring events were advanced or delayed synchronously with water addition amount increases [27]. The results do not mean that the fruiting phenology will follow the same pattern. In fact, this study found that the fruit setting and ripening time and the duration of the fruiting period were not significantly different from the control in most cases under the different water addition treatments (+25%, +50%, +75%, and +100%) and in most years (Figure 3, Figure 4 and Figure 5). The fruiting phenology remained relatively stable relative to the significant changes in leaf unfolding, leaf cessation, and even flowering after all four water addition treatments (+25%, +50%, +75%, and +100%). Similarly, rainfall was not significantly correlated with the proportion of flowers setting fruits in the Andean shrub *Befaria resinosa* (Ericaceae) [29]. A double precipitation treatment did not significantly affect the fruiting phenology in a tallgrass prairie in the south-central Great Plains in the United States [14]. A long-term change in fruiting phenology in a West African lowland tropical rainforest was also not explained by rainfall [30]. The exception of the significant advance in fruit ripening (OFR, PFR, and EFR) in 2012 (Figure 4) might have been caused by two possible reasons. First, it may relate to the extreme wet situation during the period from EFS to PFR (26.9 mm on day 178 and 32.1 mm on day 185) in 2012. The effect of water stress caused by extra water addition over a relatively extremely wet season may have resulted in the early end of fruit ripening [28]. Second, the life form can modulate the effects of precipitation on fruiting phenology [10]. In this study, the higher-magnitude advances in EFS in 2012 after water addition treatments relative to the other six years (Figure 3C) caused the significant advancing of the successional fruiting phenology. Similar results have also been reported by other authors. Accordingly, the significant advances in EFS after water addition treatments in 2012 (Figure 3C) were caused by the preceding events. In this special, extremely wet year, all phenological events, including leaf, flowering, and fruiting events, were significantly advanced [28].

### 3.2. Relative Stable Fruiting Phenology Relative to Leaf and Flower Phenology

The slight variation in fruiting events after the water addition treatments and the relatively stable response pattern with the increasing water addition amounts (Figure 6 and Figure 7) indicated that the fruiting phenology of *N. tangutorum* is independent from precipitation. Combined with the results showing that the inter-annual variation in fruiting events was not related to the temperature and precipitation factors (Table 2 and Table 3), in this study, it is reasonable to attribute this stability to inherent properties. Firstly, plant phylogenetic constraints play a pivotal role in shaping the shifts in fruiting phenology [17,31,32]. Secondly, biotic interactions could be important drivers in the fruit setting and ripening [33]. Berries of *N. tangutorum* are an important component of many food webs. For example, *Eremias multiocellata* adjust a large proportion of their diet to berries of *N. tangutorum* during hot summers to make up for the loss of surface moisture [23]. Changes in berry setting and ripening may have significant consequences for trophic interactions and ecosystem function [34,35]. Long-term coevolution has played a major role in driving the niche conservatism among species in the harsh and variable desert ecosystem. For example, in Sahelian shrubs, morphological and physiological adaptations strongly contribute to the relative independence of their activity from water availability [36]. These results indicate a strong plasticity in the fruiting phenology of *N. tangutorum* in response to precipitation increase.

### 3.3. Drivers of Inter-Annual Variations in Fruiting Phenology

The timing of fruiting phenology is the result of internal and external cues [37]. Abiotic environmental conditions such as rain [38,39,40,41,42], temperature [5,14,15,43], photoperiod, and ENSO [41,44] have been shown to play a significant role in the timing of fruiting phenology. Meanwhile, no significant correlation was determined between precipitation (winter, spring, summer, autumn, and annual) and temperature (winter, spring, summer, autumn, and annual) factors and any of the six fruiting events in this study (all *p* > 0.05, Table 2 and Table 3). We found that the inter-annual changes in fruit setting and ripening were significantly correlated with the changes in the preceding events OFL and EFL (almost all *p* < 0.05, Figure 8). In many cases, the phenological patterns have been regulated by natural selection [45]. Obviously, the strength of phylogenetic conservatism outweighed the climate effects on the fruiting phenology in this study. Taken together, our results in this study suggest that the fruiting time of *N. tangutorum* is dependent on the preceding flowering events rather than precipitation or temperature. Similar shifts have been reported in other ecosystems. For example, in one study, temporal variation in the flowering time explained 36% of the variation in the start of fruiting time [46]. The fruit setting patterns of baobabs, Adansonia digitata, were not significantly different between two years with a large rain fall difference in southern Africa [47]. A relatively stable timing of fruit setting and ripening could be associated with the formation of seeds satiating seed predators [1], which is essential for maintaining many aspects of biodiversity in a harsh and unpredictable desert environment [48].

## 4. Materials and Methods

### 4.1. Sample Area Description

Details of the experiment site, water addition treatments’ design, and the monitoring of soil moisture and soil chemical properties were reported by Bao et al. (2020, 2021). In brief, the experiment was conducted at the Desert Ecosystem Water Addition Platform (106°43′ E, 40°24′ N, 1050 m a.s.l.) in Dengkou County, Inner Mongolia, China. The annual average precipitation was 145 mm (1978–2008), with a peak wet season occurring in May– September. The annual average temperature and evaporation during that period were 7.6 °C and 2381 mm, respectively. The research site is dominated by the shrub species N. tangutorum, which often forms phytogenic nebkhas that are approximately 1~3 m high and 6~10 m in diameter with a vegetation coverage of approximately 45~75%. *N. tangutorum* nebkhas are distributed in patches on the surface of hard mud.

### 4.2. Simulated Enhancement in Precipitation

Five water addition treatments were designed to simulate precipitation increases of 0% (Control), 25% (+25%), 50% (+50%), 75% (+75%), and 100% (+100%) of the local annual average precipitation (145 mm). The water addition treatments were applied equally every month from May to September. The additional water amounts were 0, 7.3, 14.5, 21.8, and 29.0 mm each time for the five water addition treatments, respectively. The water was pumped from a well near the plots into a water tank with water meters and then transported to each sprinkler. The sprinklers, with two automatically rotating spraying arms (6 m in length) that could uniformly sprinkle water over the treatment area, were installed on the top of each nebkha (plot). More detailed information on the experimental design and the irrigation system can be found in our previous publications [27,28].

### 4.3. Phenological Observations

The phenology observations were carried out from 2012 to 2018 following the standard protocols of Phenological Observation Methodology in China [49]. Phenology recording was conducted from March 2012 to November 2018. Phenological events for all shrubs in each nebkha (plot) were recorded every other day. A series of phenological events, including leaf unfolding (onset and end), flowering (onset and end), fruit setting (onset, peak and end), fruit ripening (onset, peak, and end), and leaf cessation (leaf coloring and fall), were recorded (Figure 1, Table 4). Precipitation, air temperature, relative humidity, and evaporation data were recorded by a standard meteorological station near the experimental plots. The soil gravimetric water content (SWC) of the 0–30-cm soil layer was measured using the oven-drying method. The SWC significantly changed after the water addition treatments and was significantly linearly correlated to the water addition amounts. The influence degree of water addition on SWC was highest in spring, followed by autumn and then summer.

### 4.4. Data Processing

The observed dates of fruiting events were first transformed to day of year. The duration of the fruiting period was calculated as the difference between the EFR and OFS. The relative change of the days (∆days) was used to test the effects of the water addition treatments on each event.
(1)$$\Delta \mathrm{days}=\frac{1}{n}\sum_{i=1}^{n}\left({\mathrm{day}}_{\mathrm{treat}}-{\mathrm{day}}_{\mathrm{ctrl}}\right)$$
where day_treat_ represent the day of a given event or the duration of the fruiting period in water addition plots, day_ctrl_ represents the corresponding day in control plots, and n represents the number of the experimental years. Here, ∆days < 0 means that fruiting events (duration of fruiting period) were advanced (shortened) under water addition, while ∆days > 0 means that fruiting events (duration of fruiting period) were delayed (prolonged) under water addition.

### 4.5. Statistical Analysis

Simple linear regression analysis was used to determine the inter-annual trends of meteorological factors and fruiting events. Pearson correlation was used to analyze the relationships between fruiting events and meteorological factors. Linear mixed models were used to examine the effects of water addition treatments, year, and their interaction on fruiting events over the seven years (2012–2018). Water amount and year were used as fixed factors, while plot was used as a random factor. The dependent factor was the timing of different phenological events (Type I Sum of Squares was used). Duncan post hoc tests were used to determine pairwise differences for significant effects. One-way ANOVAs were used to test the effects of the water addition treatments on the timing of different fruiting events separately for each year. Homogeneity of variances was tested by Levine’s tests. One-sample Kolmogorov–Smirnov tests were used to validate the normality of the data distribution. All statistical analyses were completed in SPSS 20.0 (IBM, Inc.; Armonk, NY, USA), and Microsoft Excel 2019 (Redmond, WA, USA) was adopted for plotting.

## 5. Conclusions

With increasing water amounts, the fruiting phenology of N. tangutorum maintained a relatively stable response compared to the leaf and flower phenology. The fruiting phenology was found to be independent from climatic factors, precipitation, and temperature in this study. The inter-annual variations in the fruiting phenology were not driven by water or temperature but by the preceding flowering events. However, responses of fruiting phenology to environmental changes are highly species-specific [10,12,14]. More studies across species are needed in the future to test the universality of the present results in a desert ecosystem.

## Figures and Tables

**Figure 1 plants-11-01578-f001:**
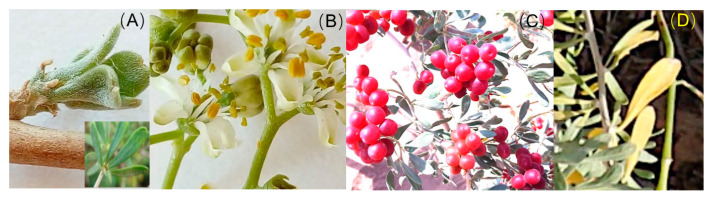
Typical phenological events of *N. tangutorum*. (**A**) Leaf unfolding, (**B**) flowering, (**C**) fruiting, and (**D**) leaf coloring.

**Figure 2 plants-11-01578-f002:**
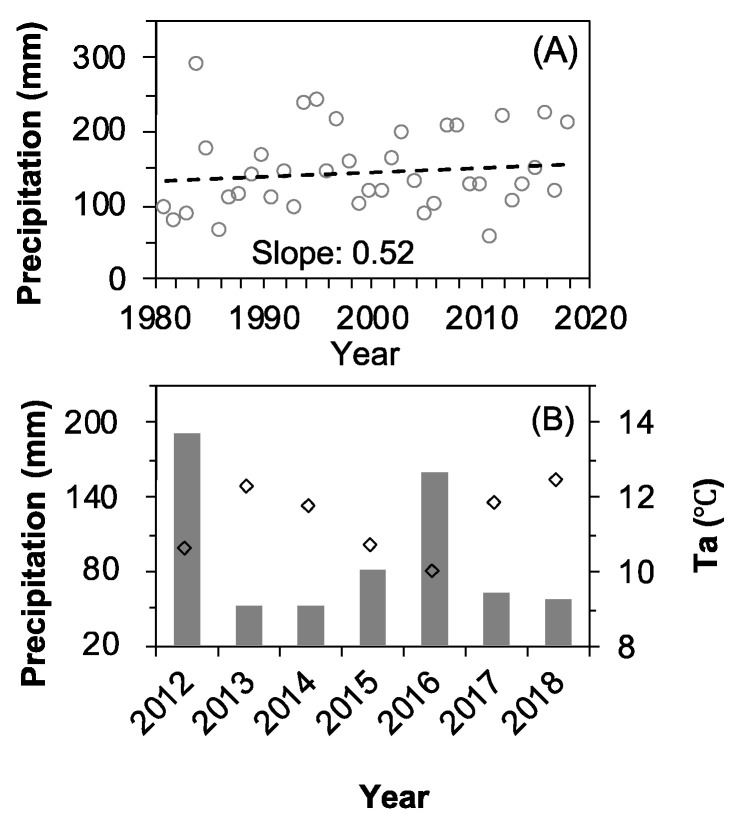
Variations in annual precipitation (**A**) and accumulated precipitation (bar) and average air temperature (diamond) from 1 January to 31 August over the period of 2012–2018 (**B**).

**Figure 3 plants-11-01578-f003:**
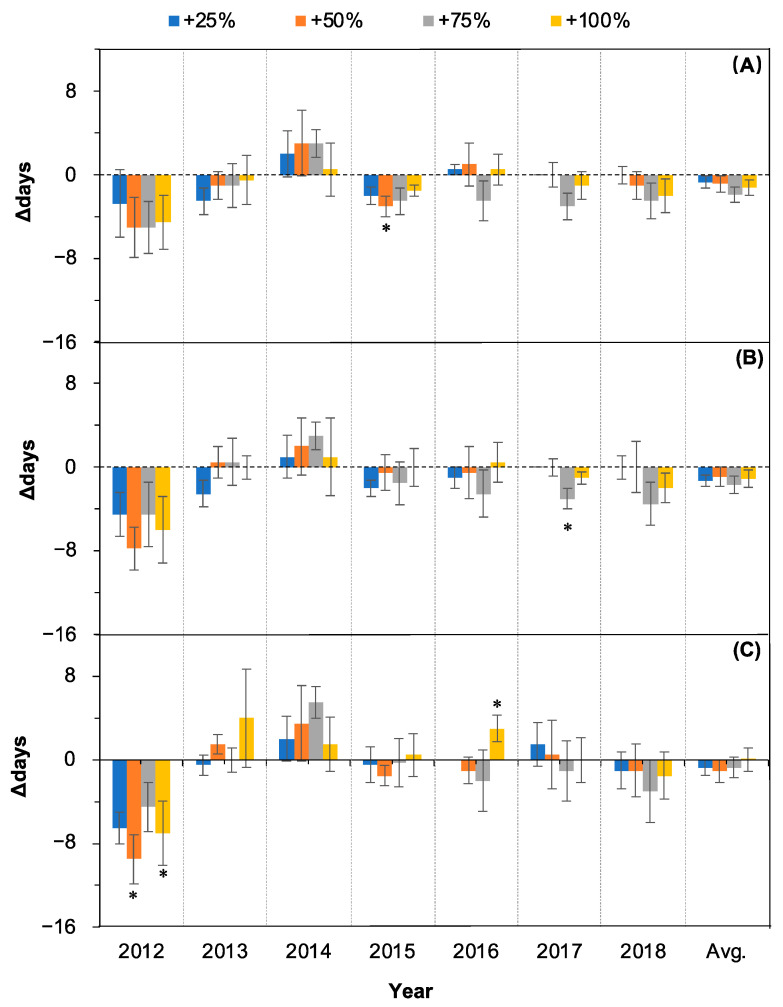
Relative changes (△days; mean ± SE) in the three fruit setting events of *Nitraria tangutorum* after water addition treatments (+25%, +50%, +75%, +100%) compared with the control. Positive and negative values represent delayed and advanced days, respectively. (**A**) OFS, (**B**) PFS, (**C**) EFS. Avg. represents average values over the period from 2012 to 2018. * Significant differences from control.

**Figure 4 plants-11-01578-f004:**
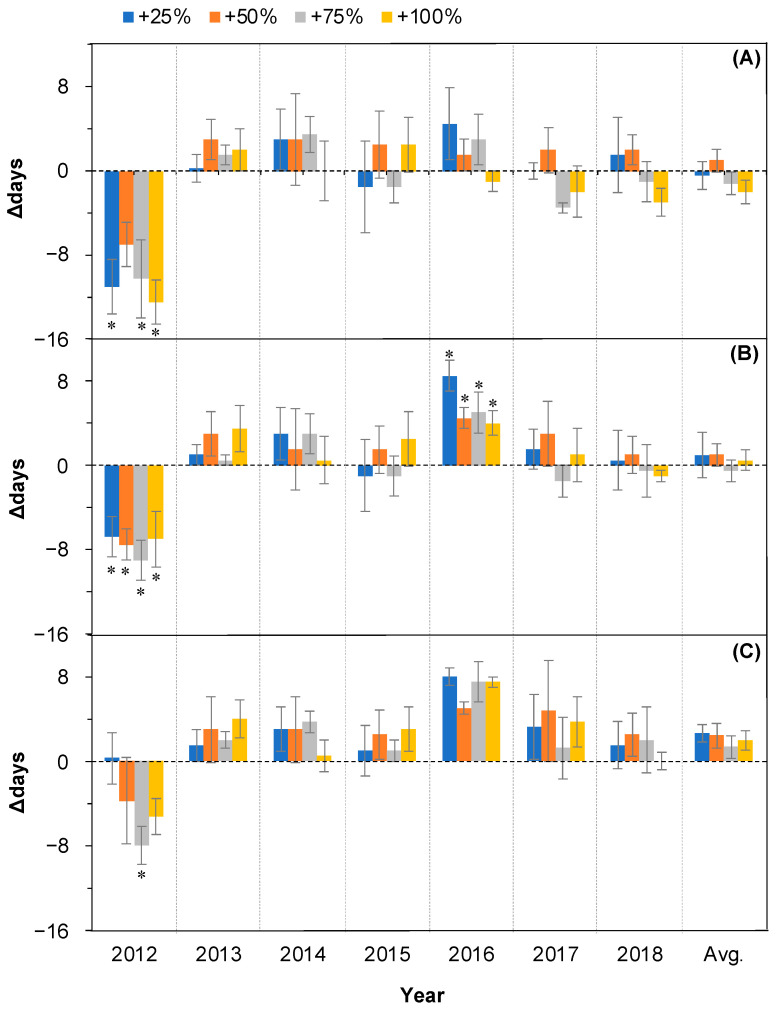
Relative changes (△days; mean ± SE) in the three fruit ripening events of *Nitraria tangutorum* after water addition treatments (+25%, +50%, +75%, +100%) compared with the control. Positive and negative values represent delayed and advanced days, respectively. (**A**) OFR, (**B**) PFR, (**C**) EFR. Avg. represents average values over the period from 2012 to 2018. * Significant differences from control.

**Figure 5 plants-11-01578-f005:**
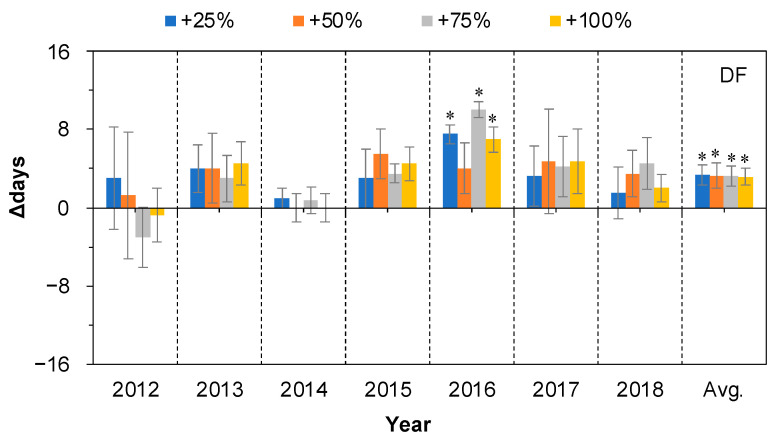
Relative changes (△days; mean ± SE) in the duration of fruiting period (DF) of *Nitraria tangutorum* after water addition treatments (+25%, +50%, +75%, +100%) compared with the control. Positive and negative values represent shortened and prolonged days, respectively. Avg. represents average values over the period from 2012 to 2018. * Significant differences from control.

**Figure 6 plants-11-01578-f006:**
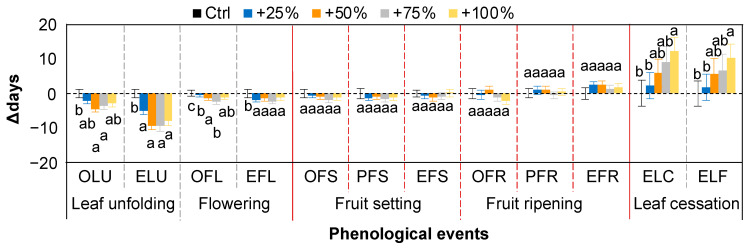
Comparison of changes (△days; mean ± SE) in phenological events of *Nitraria tangutorum* after water addition treatments (+25%, +50%, +75%, +100%) compared with the control over the period from 2012 to 2018. Fruiting events are between the solid red lines. Positive and negative values represent delayed and advanced days, respectively. OLU, onset of leaf unfolding; ELU, end of leaf unfolding; OFL, onset of flowering, EFL, end of flowering; OFS, onset of fruit setting; PFS, peak of fruit setting; EFS, end of fruit setting; OFR, onset of fruit ripening; PFR, peak of fruit ripening; EFR, end of fruit ripening; ELC, end of leaf coloring; ELF, end of leaf fall.

**Figure 7 plants-11-01578-f007:**
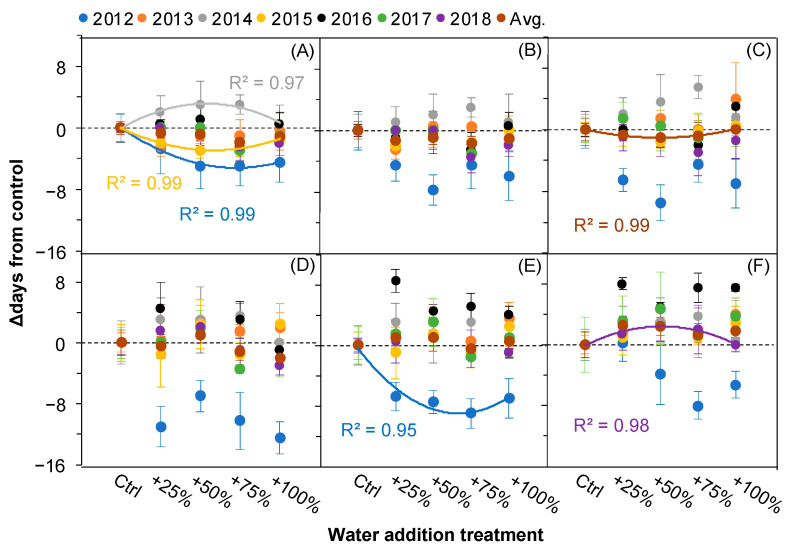
Correlations between the amount of water added (Control (Ctrl), +25, +50, +75%, +100%) and the relative changes (△days) in the fruiting events of *Nitraria tangutorum*. (**A**) OFS, onset of fruit setting; (**B**) PFS, peak of fruit setting; (**C**) EFS, end of fruit setting; (**D**) OFR, onset of fruit ripening; (**E**) PFR, peak of fruit ripening; (**F**) EFR, end of fruit ripening. Avg. represents average values over the period from 2012 to 2018. The solid lines indicate *p* ≤ 0.05.

**Figure 8 plants-11-01578-f008:**
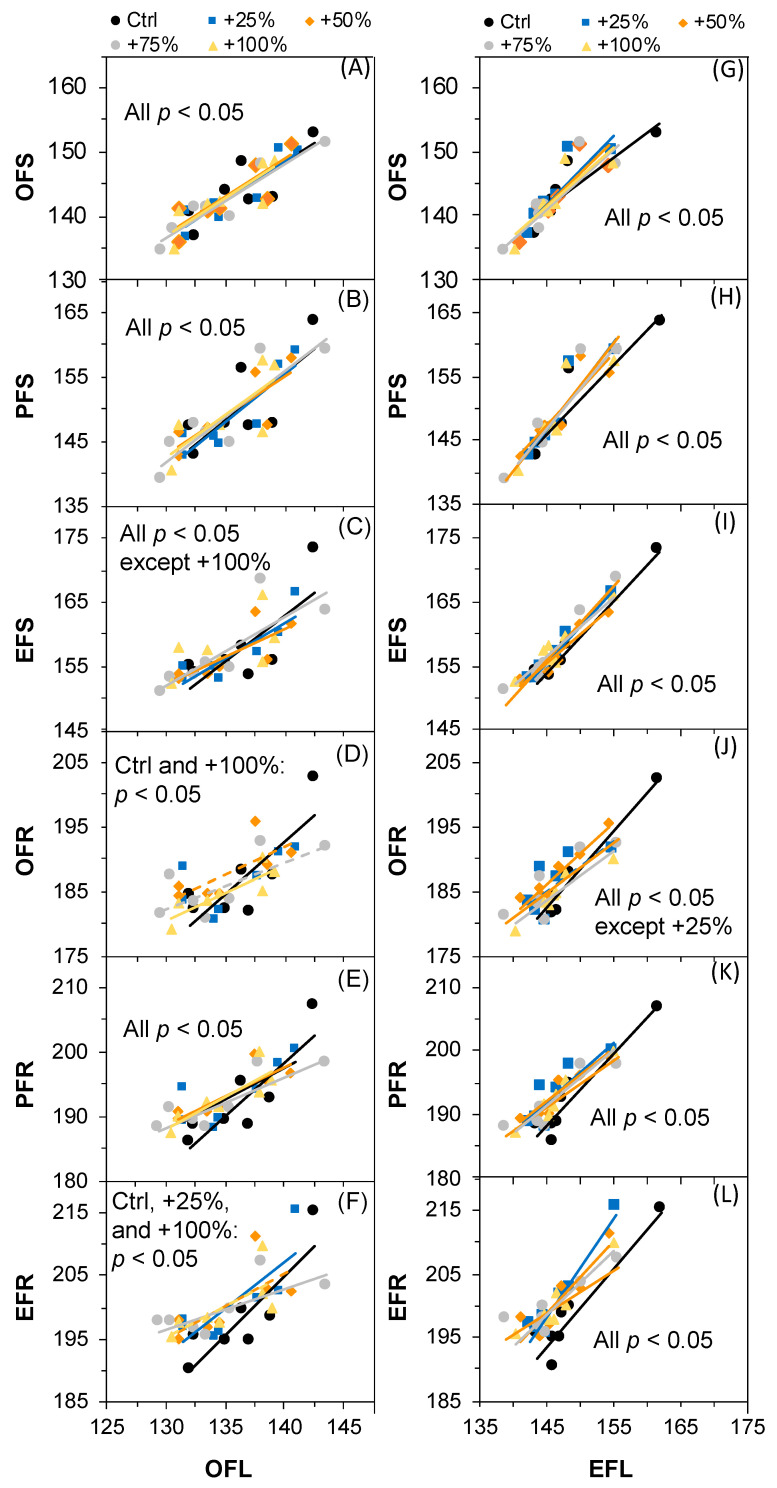
Correlations between the fruiting events and flowering events of *Nitraria tangutorum*. Correlations between OFL and OFS (**A**), PFS (**B**), EFS (**C**), OFR (**D**), PFR (**E**), EFR (**F**); and between EFL and OFS (**G**), PFS (**H**), EFS (**I**), OFR (**J**), PFR (**K**), EFR (**L**). The solid lines indicate *p* ≤ 0.05. The dashed lines indicate 0.05 ≤ *p* ≤ 0.1.

**Table 1 plants-11-01578-t001:** The *p*-values of the linear mixed model (MIXMOD) analysis on the fixed effects of water addition treatments (water), year, and their interaction on the fruiting events and the duration of the fruiting stage (DF) from 2012 to 2018. OFS, PFS, and EFS represent the onset, peak, and end of the fruit setting stage, respectively. OFR, PFR, and EFR represent the onset, peak, and end of the fruit ripening stage, respectively.

	OFS	PFS	EFS	OFR	PFR	EFR	DF
Water	0.27	0.50	0.81	0.12	0.55	0.13	0.07
Year	<0.01	<0.01	<0.01	<0.01	<0.01	<0.01	0.01
Water × Year	0.90	0.72	0.54	0.18	0.20	0.39	0.97

**Table 2 plants-11-01578-t002:** Correlations between climate factors and fruit setting events.

	OFS					PFS					EFS				
	Ctrl	+25%	+50%	+75%	+100%	Ctrl	+25%	+50%	+75%	+100%	Ctrl	+25%	+50%	+75%	+100%
AMP	0.40	0.49	0.75	0.82	0.59	0.27	0.40	0.66	0.57	0.43	0.66	0.55	0.30	0.43	0.59
AMT	0.37	0.39	0.43	0.37	0.52	0.37	0.32	0.36	0.38	0.53	0.38	0.43	0.55	0.45	0.22
T_Win_	0.44	0.65	0.81	0.77	0.69	0.24	0.42	0.62	0.45	0.47	−0.68	−0.48	−0.41	−0.44	−0.66
T_Spr_	0.45	0.46	0.48	0.59	0.33	0.59	0.63	0.61	0.63	0.43	−0.15	−0.25	−0.11	−0.21	−0.31
T_Sum_	0.64	0.66	0.72	0.72	0.53	0.75	0.82	0.78	0.75	0.57	−0.08	−0.11	0.02	−0.13	−0.22
T_Aut_	0.66	0.44	0.20	0.25	0.27	0.89	0.67	0.34	0.46	0.44	−0.33	−0.06	0.17	0.09	−0.01
P_Win_	0.91	0.89	0.81	0.90	0.97	0.80	0.99	0.96	0.76	0.86	−0.17	0.03	−0.01	−0.12	−0.22
P_Spr_	0.41	0.61	0.75	0.48	0.48	0.66	0.78	0.69	0.61	0.60	0.17	0.15	0.10	0.26	0.14
P_Sum_	0.44	0.54	0.71	0.87	0.60	0.25	0.39	0.65	0.55	0.43	0.67	0.55	0.35	0.41	0.67
P_Aut_	0.76	0.92	0.80	0.93	0.80	0.67	0.80	0.83	0.76	0.81	−0.22	−0.17	−0.23	−0.10	−0.35

**Table 3 plants-11-01578-t003:** Correlations between climate factors and fruit ripening events.

	OFR					PFR					EFR				
	Ctrl	+25%	+50%	+75%	+100%	Ctrl	+25%	+50%	+75%	+100%	Ctrl	+25%	+50%	+75%	+100%
AMP	0.59	0.43	0.41	0.44	0.34	0.33	0.31	0.61	0.46	0.47	0.36	0.21	0.49	0.38	0.30
AMT	0.45	0.26	0.55	0.26	0.35	0.15	0.47	0.12	0.34	0.25	0.12	0.25	0.08	0.18	0.24
T_Win_	−0.61	−0.39	−0.40	−0.51	−0.24	0.24	0.33	0.55	0.40	0.42	0.20	0.16	0.36	0.25	0.25
T_Spr_	−0.15	−0.17	−0.18	−0.19	−0.46	0.91	0.62	0.73	0.75	0.53	0.94	0.84	1.00	0.98	0.65
T_Sum_	0.00	−0.02	0.02	−0.10	−0.27	0.85	0.92	0.83	0.94	0.87	0.74	0.90	0.66	0.79	0.96
T_Aut_	−0.23	0.11	0.05	0.18	0.35	0.71	0.78	0.70	0.62	0.70	0.56	0.63	0.78	0.92	0.81
P_Win_	0.01	0.21	0.18	−0.06	0.10	0.99	0.71	0.53	0.95	0.84	0.93	0.93	0.72	0.91	0.75
P_Spr_	0.04	−0.35	0.02	−0.16	0.33	0.75	0.64	0.95	0.92	0.62	0.76	0.97	0.90	0.86	0.86
P_Sum_	0.65	0.57	0.47	0.56	0.33	0.31	0.17	0.48	0.36	0.40	0.33	0.14	0.41	0.27	0.20
P_Aut_	−0.31	−0.26	−0.28	−0.27	−0.17	0.59	0.49	0.46	0.68	0.50	0.56	0.47	0.49	0.56	0.33

**Table 4 plants-11-01578-t004:** Phenological observation and observation methods of *N*. *tangutorum*.

Phenological Events		Observation Methods
Leaf unfolding	Onset (OLU)	At least one young leaf has completely extended and spread out completely from one or more leaf buds observed on the whole nebkha.
	End (ELU)	More than 90% of the young leaves on leaf buds of the whole nebkha have been completely spread.
Flowering	Onset (OFL)	At least one flower bud on the whole nebkha has fully opened.
	End (EFL)	More than 90% of the flower buds in each nebkha have fully opened.
Fruit setting	Onset (OFS)	At least one flower on the whole nebkha has at least one green berry visible.
	Peak (PFS)	More than 50% of flowers on the whole nebkha have their green berries developed.
	End (EFS)	More than 90% of flowers on the whole nebkha have their green berries developed.
Fruit ripening	Onset (OFR)	At least one berry on the whole nebkha is developing a red color.
	Peak (PFR)	More than 50% of berries on the whole nebkha are developing red colors.
	End (EFR)	More than 90% of berries on the whole nebkha are developing red colors.
Leaf cessation	End of leaf coloring (ELC)	More than 90% of the leaves on the whole nebkha have turned yellow.
	End of leaf fall (ELF)	More than 90% of the leaves on the whole nebkha have fallen off.

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
