# Peer review of "Preceding Phenological Events Rather than Climate Drive the Variations in Fruiting Phenology in the Desert Shrub Nitraria tangutorum"

_plants, 2022, doi:10.3390/plants11121578_

Round 1

Reviewer 1 Report

The manuscript plants-1758617 entitled ‘Preceding phenological events rather than climate drive the variations of fruiting phenology in a desert shrub Nitraria tangutorum’ is a well-written manuscript presenting studies on the water impact on fruiting phenology.

Used methods allow to get interesting results and thus allow to supply the manuscript with high-quality figures. However, the results and conclusions obtained are very intriguing. The authors showed that water addition treatments had no significant effects on the onset, peak, and end time of fruit ripening. Also, the authors showed that water addition treatments had no significant effects on the duration of the fruiting period.

 I found small mistakes in the text:

Whole manuscript: Nitraria tangutorum, as well as other species, should be written in italics.

Line 60: „natural’

Line 318: „figure’

 In my opinion, it will be a good idea to support this conclusion with genetic studies about changes or lack of changes in gene expression as a result of water treatment. As well, microscopic embryological studies of every phenological event will be interesting.

Overall, due to the very significant importance of agricultural research to better understanding physiological mechanisms, and the difficulty of used methods, the manuscript should be accepted for publication. My comments are only editorial, therefore, in my opinion, the manuscript can be accepted for publication in its current form.

Author Response

Point 1: The manuscript plants-1758617 entitled ‘Preceding phenological events rather than climate drive the variations of fruiting phenology in a desert shrub Nitraria tangutorum’ is a well-written manuscript presenting studies on the water impact on fruiting phenology.

Used methods allow to get interesting results and thus allow to supply the manuscript with high-quality figures. However, the results and conclusions obtained are very intriguing. The authors showed that water addition treatments had no significant effects on the onset, peak, and end time of fruit ripening. Also, the authors showed that water addition treatments had no significant effects on the duration of the fruiting period.

Response: We really appreciate your kindly approbatory comments on our manuscript.

Point 2:  I found small mistakes in the text:

Whole manuscript: Nitraria tangutorum, as well as other species, should be written in italics.

Line 60: „natural’

Line 318: „figure’

Response: Thanks very much for the correction. All these errors have been fixed in the revised manuscript.

Point 3:   In my opinion, it will be a good idea to support this conclusion with genetic studies about changes or lack of changes in gene expression as a result of water treatment. As well, microscopic embryological studies of every phenological event will be interesting. Overall, due to the very significant importance of agricultural research to better understanding physiological mechanisms, and the difficulty of used methods, the manuscript should be accepted for publication. My comments are only editorial, therefore, in my opinion, the manuscript can be accepted for publication in its current form.

Response: Again, we reaaally appreciate your suggestive comments to our future studies and projects. I think we will do some gene scale or at least some microscopic embryological studies in the following years.

Reviewer 2 Report

Introduction
This part could be implemented with the addition of some bibliographical references (i.e. about water effects on ecology and phenology) as descripted below in R&D.

Material and Methods
it would be more useful to describe the methodology rather than refer to other publications (Line 384-385)

Results and Discussions
the results and discussions are clear and consistent. The figures are not always understandable and should be enlarged

Conclusions
In line with the manuscript goal, should be improved.

Author Response

Point 1: Introduction
This part could be implemented with the addition of some bibliographical references (i.e. about water effects on ecology and phenology) as descripted below in R&D.

Response: Thanks very much for your comments. We did have read several typical books (as listed bellow for example) related to the phenology and climate change. Most of their chapters, bring us a lot of inspirations, focused on the relationships between temperature and phenology. However, less attention have been paid to the effects of water change on plant phenology. The contents of those books are not relevant to this manuscript. As a result, we have to keep the introduction in the current form. Thanks again.

Mark D. Schwartz (ed.), 2013, Phenology: An Integrative Environmental Science (2nd Edition). Dordrecht, The Netherlands: Springer, DOI 10.1007/978-94-007-6925-0

Xiaoqiu Chen (ed.), 2017, Spatiotemporal Processes of Plant Phenology: Simulation and Prediction. Springer, Berlin, Heidelberg. DOI: 10.1007/978-3-662-49839-2

Gunther Schmidt; Simon Schönrock; Winfried Schröder (ed.), 2014, Plant Phenology as a Biomonitor for Climate Change in Germany, Springer, Cham. DOI: 10.1007/978-3-319-09090-0

Asko Noormets (ed.), 2009, Phenology of Ecosystem Processes. Springer, New York, NY DOI: 10.1007/978-1-4419-0026-5

Helmut Lieth (ed.), 1974, Phenology and Seasonality Modeling. Springer, Berlin, Heidelberg. DOI: 10.1007/978-3-642-51863-8

Xiaoyang Zhang (ed.), 2012, Phenology and Climate Change. INTECH Open Access Publisher. ISBN 978-953-51-0336-3

Point 2: Material and Methods
it would be more useful to describe the methodology rather than refer to other publications (Line 384-385)

Response: Thanks very much for the comments. This has been rewritten. Please see details in the  revised manuscript.

Point 3: Results and Discussions
the results and discussions are clear and consistent. The figures are not always understandable and should be enlarged

Response: Thanks very much for your kindly comments. All figures have been enlarged greatly. Please see details in the  revised manuscript.

Point 4: Conclusions
In line with the manuscript goal, should be improved.

Response: Thanks very much for your kindly comments. The coclusion has been rewrite. Please see details in the revised manuscript.
